# Comparative Analysis of Human and Animal *E.* *coli*: Serotyping, Antimicrobial Resistance, and Virulence Gene Profiling

**DOI:** 10.3390/antibiotics11050552

**Published:** 2022-04-21

**Authors:** Mahmoud M. Bendary, Marwa I. Abdel-Hamid, Walaa A. Alshareef, Hanan M. Alshareef, Rasha A. Mosbah, Nasreen N. Omar, Mohammad M. Al-Sanea, Majid Alhomrani, Abdulhakeem S. Alamri, Walaa H. Moustafa

**Affiliations:** 1Department of Microbiology and Immunology, Faculty of Pharmacy, Port Said University, Port Said 42511, Egypt; 2Department of Microbiology, Faculty of Veterinary Medicine, Zagazig University, Zagazig 44511, Egypt; mero_micro2006@yahoo.com; 3Department of Microbiology and Immunology, Faculty of Pharmacy, October 6 University, 6 October 12566, Egypt; lolowation@gmail.com; 4Department of Pharmacy Practice, Faculty of Pharmacy, University of Tabuk, Tabuk 71491, Tabuk, Saudi Arabia; halsharef@ut.edu.sa; 5Infection Control Unit, Zagazig University Hospital, Zagazig 44511, Egypt; rashamosbah1@yahoo.com; 6Department of Biochemistry, Faculty of Pharmacy, Modern University for Technology and Information, Cairo 19448, Egypt; nasreen.nabil@pharm.mti.edu.eg; 7Pharmaceutical Chemistry Department, College of Pharmacy, Jouf University, Sakaka 72341, Al-Jouf, Saudi Arabia; 8Department of Clinical Laboratories Sciences, The Faculty of Applied Medical Science, Taif University, Taif 26432, Makkah, Saudi Arabia; m.alhomrani@tu.edu.sa (M.A.); a.alamri@tu.edu.sa (A.S.A.); 9Centre of Biomedical Science Research (CBSR), Deanship of Scientific Research, Taif University, Taif 26432, Makkah, Saudi Arabia; 10Microbiology and immunology Department, Faculty of Pharmacy, Helwan University, Cairo 19448, Egypt; walaa_hassan@pharm.helwan.edu.eg

**Keywords:** *E. coli*, MDR, multi-virulent, serotypes, genetic diversity

## Abstract

Widespread multidrug-resistant (MDR) and multi-virulent diarrheagenic *E. coli* create several crises among human and animal populations worldwide. For this reason, we looked forward to a breakthrough with this issue and tried to highlight these emerging threats. A total of 140 diarrheagenic *E. coli* isolates were recovered from animal and human sources. The O26 serotype, alongside the ampicillin/cefoxitin resistance phenotype, was predominant among both human and animal isolates. Of note, imipenem represented the most effective antibiotic against all the investigated isolates. Unfortunately, 90% and 57.9% of the tested isolates showed MDR and multi-virulent patterns, respectively. The animal isolates were more virulent and showed higher sensitivity to antimicrobial agents. Both animal and human isolates could not be arranged into related clusters. A strong negative correlation between the existence of virulence genes and antimicrobial resistance was clearly detected. A significant correlation between serotypes and antimicrobial resistance was not detected; meanwhile, a significant positive correlation between some serotypes and the presence of certain virulence genes was announced. Finally, our results confirmed the urgent need for restricted guidelines, in addition to new alternative therapies, due to the genetic diversity and wide spreading of MDR side by side with multi-virulent *E. coli* isolates.

## 1. Introduction

High morbidity and mortality rates have been detected in the developing world, due to diarrhea, which receives more attention in these countries. Several new phenotypes of enteric pathogens were recorded, which cause severe gastroenteritis [1]. One of the most important etiologic agents of diarrhea is *E. coli*. Although *E. coli* is normal commensal flora, it may convert to a full pathogen by the acquisition of mobile genetic materials, through horizontal gene transfer, or bacteriophages, causing several intestinal and extraintestinal infections, which affect both humans and animals [2].

*Escherichia coli* (*E. coli*), as a widespread bacterial pathogen, has adaptive ability in diverse ecological niches. *E. coli* strains could potentially have a serious impact on both humans and animals, including cattle, horses, and birds, since they can induce enteric [3] and extraintestinal infections [4]. Moreover, *E. coli* is involved in urinary tract infections, sepsis/meningitis and gastrointestinal diseases. There are two major public health crises in terms of *E. coli* infections. Firstly, widespread pathogenic *E. coli* in environmental water, such as irrigation and drinking water, causes severe infections in humans and animals [5]. Secondly, the presence of multi-virulent and multidrug-resistant (MDR) *E. coli* strains, which exhibited resistance to at least three antimicrobial drugs from different classes [6]. 

According to the disease type, *E. coli* can be classified into two main types: diarrhoeagenic *E. coli* (DEC) and extraintestinal pathogenic *E. coli* (ExPEC). The ExPEC has two pathovars: uropathogenic *E. coli* and neonatal meningitis *E. coli*. Moreover, there are six pathovars of DEC, according to the mechanisms of diarrhea, including enteroinvasive *E. coli*, enteropathogenic *E. coli*, enteroaggregative *E. coli*, diffusely adherent *E. coli*, enterotoxigenic *E. coli*, and enterohaemorrhagic *E. coli* [2]. The genetic background of these pathovars is diverse, and their evolution can occur through the acquisition and loss of pathogenicity islands [7]. Many virulence genes of recognized importance are associated with severe infections caused by *E. coli* strains. They include genes encoding for adhesion (*fim*H and *kps*M), enterotoxin (*ast*A), intimin (*eae*A), Vero toxins (*vtx*), Shiga toxins (*stx*), hemolysin (*hly*) and outer membrane proteins (*omp*). Additionally, transcriptional activators for aggregative adherence fimbriae and invasion were encoded by *agg*R and *invE* genes, respectively [8,9,10]. These virulence factors are involved in the pathogenicity of *E. coli* strains, as they help these organisms to adhere to the host surfaces, invade host cells and tissues, avoid host defense mechanisms, and incite noxious inflammatory responses, thereby causing clinical diseases [11]. 

Regarding the fact that DEC is widespread throughout the world, high prevalence rates of DEC were found among childhood diarrhea in Kenya (55.9%) [12], Peruvian (43.0%), coastal periurban areas of Peru (31.5%) [13], and rural South Africa (40.8%) [14]. Moreover, moderate prevalence rates were found in Mexico (23%) [15], Vietnam (22.5%) [16], and Indonesia (21%) [12]. On the other hand, the prevalence rate was low in Tanzania (10%) [17]. The variation in prevalence rates may be attributed to climate change adaptation, as the prevalence rate increases by 5% when the temperature increases by one degree [18]. 

The role of antibiotic misuse in developing the resistance among *E. coli* strains cannot be ignored. It is caused by the absence of restricted regulation of the administration of antimicrobials, in addition to some bad habits, such as unwarranted and self-prescription [19]. The widespread resistance among *E. coli* strains might be a result of soil, aquatic, and environmental contamination by untreated manure or sewage coming from animals or humans treated with several antibiotics [20], as most antimicrobial agents are not fully eliminated during the treatment process of sewage [21]. Therefore, soil and aquatic environments could facilitate the dissemination of resistance genes among *E. coli* strains [22], especially in developing countries. 

It is noteworthy that the possibility of the simultaneous transmission of virulence genes with resistance genes in mobile genetic materials may induce the emergence of new pathogenic strains [23]. The infections with these new strains are particularly problematic, due to the lack of susceptibility to antibiotic medications and the extraordinary ability to express a large repertoire of virulence genes. Therefore, next-generation therapies and innovative drug repurposing are urgently needed [24,25,26]. In light of the above-mentioned threats, the present study was designed to elucidate the phylogenetic analysis depending on phenotypic and genotypic characterization of MDR and multi-virulent diarrheagenic *E. coli* (DEC) isolates from several sources, to gain a clear picture of the epidemic situation of these strains in Egypt. 

## 2. Results

### 2.1. Phenotypic Characterization of DEC

Out of 350 samples collected from diarrhetic horses, cows, and humans, 140 (40%) DEC isolates were detected on the basis of their biochemical and cultural properties. The results of API 20E system and PCR amplification of the 16SrRNA gene were in agreement with those of conventional methods. The highest prevalence rate of DEC (53.3%, 80/150) was recorded among human samples. A sum of 35 and 25 DEC isolates was recovered from cow and horse diarrheal stool specimens, with isolation percentages of 35 and 25%, respectively. The results of the serogroup analysis using specific polyvalent and monovalent O antisera revealed seven different serotypes (O26, O44, O55, O151, O125, O145 and O1) among 119 *E. coli* isolates; meanwhile, 21 isolates (15%) were untypeable. Notably, *E. coli* O26 was the most predominant serotype (25%, 35/140) distinguished from the typable strains. As for the distribution of other serotypes, both O44 and O55 were detected with a high frequency (15% each), followed by O1 and O125 serotypes (10% each). On the other hand, serotypes O151 and O145 were detected with a low percentage (5% each, 7/140). The distribution of various serotypes represented among the 140 DEC isolates is shown in Figure 1, Figure 2A and Appendix A.

### 2.2. Antimicrobial Susceptibility Patterns

The antimicrobial susceptibility patterns of all the recovered DEC isolates revealed that the highest resistance rates were shown in ampicillin, followed by cefoxitin (89.3 and 85.7%, respectively). On the other hand, imipenem represented the most effective antibiotic against many of the DEC isolates (59.9%), followed by ciprofloxacin (46.4%) (Figure 1 and Appendix A). The analyzed data demonstrated that there were significant differences in the resistance patterns of most of the recovered DEC isolates towards different antimicrobials (*p* < 0.005). Unreasonably, 90% (126/140) of the investigated *E. coli* isolates showed the MDR patterns. Taken together, an analysis of our data provided further evidence of the high prevalence of antimicrobial resistance among human isolates, as illustrated in Figure 2B. Confirming this context, 3.7 and 21.7% of the human and animal isolates showed the non-MDR patterns, respectively. Surprisingly, both the human and animal isolates could be treated with imipenem, ciprofloxacin and piperacillin/tazobactam, as shown in Figure 2B. The multiple antibiotic resistance (MAR) index of the isolates ranged from 0.07 to 1 (resistant to all the tested antimicrobials). Five out of eight isolates with an MAR index of one belonged to human isolates. It was demonstrated that the MAR index of the antimicrobials varied from 0.030 to 0.064. The MAR index was found to be the highest for ampicillin (0.064), in contrast to imipenem (0.030).

### 2.3. Virulence Gene Profiles

Concerning PCR determination of DEC virulence genes, both *ompA* and *fimH* genes were detected in all the tested isolates, but none of these isolates carried *kps*MTII, *hly* and *vt2e* genes (Figure 1 and Figure 2C). Most commonly, the *astA* gene was detected among 43.6% of the DEC isolates; meanwhile, only 11 isolates (7.9%) harbored the *invE* gene. The distribution of virulence genes among human and animal DEC isolates is illustrated in Figure 2C and Appendix A. Interestingly, the virulence genes were identified in animal isolates more often than in human isolates. The majority of the human and animal isolates were categorized, depending on the presence of virulence genes, as multi-virulent, as well as being MDR (Figure 2D).

### 2.4. Correlation between Serotypes, Existence of Virulence Genes and Antimicrobial Resistance

Regarding the correlation between DEC serotypes and their virulence genes, the existence of *invE* and *stx*2 genes was observed to be positively correlated with the O145 serotype (R = 0.27 and 0.41, respectively). Meanwhile, their frequency was noted to have a slight negative correlation with the O44 serotype, as shown in Figure 3A and Appendix A. Interestingly, a significant negative correlation (R = −0.5:−1, *p* < 0.05) between antimicrobial resistance and the existence of virulence genes was observed (Figure 3B and Appendix A). The correlation among antimicrobial resistance revealed an overall significant positive correlation (R = 0.5:1, *p* < 0.05) among ampicillin, amoxicillin/clavulanic acid, aztreonam, cefoxitin, cefoperazone, and cefepime, as observed in Figure 3B,C. According to Figure 3C and Appendix A, there was no correlation between the DEC serotypes and their antimicrobial resistance.

Based on serotyping, antimicrobial resistance and virulence gene profiles, all the tested DEC isolates (140), with the exception of 14 isolates (3 pairs of human and equine isolates, 2 pairs of human isolates, and 2 pairs of human and cow isolates), could be typed into different lineages, as shown in Figure 4.

## 3. Discussion

One of the most important worldwide crises is the wide spreading of infections with resistant microorganisms, especially multidrug-resistant (MDR) bacterial and fungal pathogens [23,27,28,29,30]. These crises were compounded by the evolution of both microbial resistance and virulence [31]. Recently, many reports have announced the increase in foodborne infections, which are attributed to resistant animal pathogens with multi-virulence arrays [32,33,34]. In this context, many health-associated problems were linked to DEC, which can be transmitted through several food chains of animal origin. Unfortunately, there were not enough details about the phylogenetic analysis of animal and human DEC. Scarce reports were available regarding the correlation analysis within and among different DEC serotypes, especially in Egypt. Therefore, this study has focused on the great emergence and heterogeneity of MDR/multi-virulent DEC serotypes among human and animal hosts.

The occurrence of *E. coli* was recorded in several reports among different countries, depending on the disparity in sample types, sampling schemes, geographic location, detection protocols, and seasonal and environmental factors. In this study, the prevalence rate of *E. coli* (40%) was in accordance with that observed in Bangladesh (41%) [35]; however, it was much higher in comparison with those recorded in several studies carried out in Egypt (23%) [36] and Zimbabwe (20%) [37]. Regarding the serotyping results, seven different serotypes, O26, O44, O55, O151, O125, O145 and O1, were detected, with the predominance of the O26 serotype (25%). Of note, there is a variation in the serotypes of *E. coli* among different studies worldwide [30,38]. This variation can be accepted on the basis of the positive correlation between *E. coli* serotypes and the types of diseases or clinical forms of their infections.

Studying the epidemiology, virulence, and antimicrobial resistance of *E. coli* serotypes, in addition to their correlation, is essential to improve the control and treatment of this pathogen. In this context, and based on the antimicrobial susceptibility results of *E. coli*, the maximum antimicrobial resistance was recorded towards ampicillin and cefoxitin. In other studies, higher resistance rates have been announced to tetracycline in Brazil [39], Egypt [37], and Korea [40]. Meanwhile, in India [41] and Egypt [38], higher prevalence rates of *E. coli* resistance were detected to erythromycin and streptomycin. These variations in antimicrobial resistance may be associated with the type of prescribed antimicrobial agents in various hospitals, alongside different geographical areas. Additionally, the antimicrobial usage in animal production and the variations in legislation that guide the use of antibiotics from one region to another may lead to these variations [42]. Unfortunately, the current study indicated an alarmingly high prevalence of MDR (90%) among the tested *E. coli* strains, in addition to their high MAR indices, especially for the human isolates. Therefore, continuous surveillance for antimicrobial resistance among *E. coli* strains is urgently needed to avoid treatment failure.

Virulence factors and antimicrobial resistance are necessary to overcome the host immune response and enhance the ability of bacteria to survive under adverse conditions [43]. The presence of multiple virulence factors, as observed in our isolates, may exacerbate this issue. Alarmingly, the vast majority of our human and animal isolates were MDR, as well as being multi-virulent. This reflects the evolution of the epidemic situation in Egypt. Our results showed high diversity among, and within, animal and human DEC isolates. There were no similarities between the profiles of human and animal strains, as they were clustered in different lineages. Therefore, the findings of this manuscript did not support the possibility of zoonotic transmission among our isolates. The lack of host adaptation, in addition to the chance of contamination between both hosts and the heterogeneous nature of the *E. coli* genome, may have led to this result [42].

The correlation between antimicrobial resistance and virulence production is difficult to understand, due to the complex phylogenetic background of the bacteria, which depends on several factors, such as the bacterial species, host, ecological niche, and virulence and resistance mechanisms [43]. We observed a strong negative correlation between antimicrobial resistance and virulence gene profiles. Several previous studies announced that the acquisition of antibiotic resistance leads to a fitness cost [44,45]. These results strengthen the hypothesis that the loss of pathogenic fitness and virulence potential has always been associated with the acquisition of antimicrobial resistance [46]. In the same context, there was no absolute correlation between *E. coli* serotypes and antimicrobial resistance. Meanwhile, a positive correlation between some serotypes and the existence of certain virulence factors was observed. Consistent with our results, several reports announced that there was an association between *E. coli* serotypes and some virulence genes [47,48]. Moreover, another study demonstrated the diversity of virulence genes and serotypes among *E. coli* strains [48]. In contrast to our findings, several studies found a significant relationship between certain microbial serotypes and antimicrobial resistance [49,50,51]. This indicates that there is no fixed or absolute correlation between serotypes and antimicrobial resistance and virulence production. Furthermore, there is high genetic variation within each *E. coli* serotype, as the strains within a single serotype belong to different lineages or genotypes. This supports the evidence that there is no single factor responsible for the virulence and resistance of *E. coli* strains, and their expression may be affected by several environmental conditions.

## 4. Materials and Methods

### 4.1. Study Design and DEC Isolation and Identification

This is a retrospective study that was conducted on 350 stool samples collected from diarrhetic animals (horses (100) and cows (100)) and humans (150) from both university and private hospitals in Port Said and Sharkia Governorates. All the collected samples were initially inoculated into brain heart infusion broth (Oxoid, UK) to enhance the growth of *E. coli*. A loopful from the cultured broth was inoculated into MacConkey and eosin methylene blue agar media (Oxoid, UK), and then the plates were incubated at 37 °C for 24 h. Suspected *E. coli* isolates were identified according to their cultural and biochemical characteristics, such as citrate utilization, indole production and reactions on triple sugar iron agar medium (Oxoid, UK) [52]. Confirmation of *E. coli* isolates was performed using API 20E strips (BioMérieux, Mary l’Etoile, France) and specific polymerase chain reaction (PCR) amplification of the *16S rRNA* gene [53]. The following cycle condition was used to amplify ECP79F-ECR620R regions of the *16S rRNA* gene: pre-heating at 95 °C for 5 min, then forty cycles of 94 °C for 45 s, 55 °C for 45 s, and 72 °C for 1.5 min, and a final extension for 5 min at 72 °C. *E. coli* ATCC 25,922 and *Staphylococcus aureus* ATCC 25,923 were used as positive and negative controls, respectively.

### 4.2. Serotyping

The serotyping of DEC strains was performed by an agglutination test using polyvalent and monovalent O-specific antisera (Test Sera Enteroclon, Anti–Coli, Germany), according to the manufacturer’s recommendations. Briefly, a clean glass slide was divided into 2 sections by a marker pin, and then one drop of polyvalent O-specific antisera (test) and one drop of physiological saline (control) were added to each section. One drop of DEC suspension was mixed with the test and control drops. The slide was tilted and the agglutination was recorded. This test was repeated using monovalent O-specific antisera instead of the polyvalent serum, which agglutinated.

### 4.3. Antimicrobial Susceptibility Testing

In vitro determination of the susceptibility patterns of all the recovered DEC isolates to various antimicrobials was conducted, adopting the Kirby–Bauer disc diffusion method using Mueller–Hinton agar and standard antimicrobial discs, according to the Clinical and Laboratory Standards Institute (CLSI) recommendations [54]. The antimicrobial agents tested were ampicillin (AMP), cefoperazone (CPZ), cefoxitin (FOX), amoxicillin/clavulanic acid (AMC), aztreonam (ATM), piperacillin/tazobactam (TZP), chloramphenicol (C), imipenem (IPM), gentamycin (CN), ciprofloxacin (CIP), sulfamethoxazole/trimethoprim (SXT), erythromycin (E), cefepime (CPM), and tetracycline (TE). Susceptibility to antimicrobials using the broth microdilution method [55] was also determined for all DEC isolates, to check the results of disc susceptibility testing. The isolates that showed non-susceptibility patterns to at least one agent in three or more different classes of antimicrobial agents were defined as MDR [56].

### 4.4. Determination of Multiple Antibiotic Resistance (MAR) Indices

The multiple antibiotic resistance (MAR) index is one of the tools that reveals the spread of resistant pathogens in a given population. It is a good way to identify the relative risk for the source of contamination, as the multiple and abuse of the antibiotic were recorded when the MAR was >0.2 [1]. The MAR index values for each isolate and antimicrobial were calculated [57,58] according to the following formulas: MAR index for the isolate = number of antimicrobials to which the isolate was resistant/number of tested antimicrobials. MAR index for the antimicrobial = number of antimicrobial-resistant isolates/number of antimicrobials x number of isolates.

### 4.5. Virulence Genotyping of E. coli

The total bacterial genomic DNA was isolated from DEC isolates using QIAamp DNA Mini Kit (QIAGEN GmbH, Hilden, Germany). Specific PCR assays were then performed to screen for the presence of some virulence genes using specific primers and thermal cycling conditions described elsewhere [59,60,61,62,63,64,65]. The gene-specific primer sequences, as well as the amplicon sizes, are listed in Table 1. In each PCR run, appropriate positive and negative controls were included. The amplicons were then analyzed by agarose gel electrophoresis.

### 4.6. Statistical Analysis

The antimicrobial susceptibility data were expressed as percentages. The significant differences in the levels of resistance among the selected antimicrobials were determined depending on two-way analysis of variance (ANOVA), without replication. The P value of <0.05 was considered to be statistically significant. All statistical and correlation analyses were carried out using the R package corrplot, heatmaply, hmisc, ggpubr and GraphPad Prism (version 6; GraphPad Software Inc.; San Diego, CA, USA).

## 5. Conclusions

The heterogeneity of our DEC isolates among both human and animal hosts was an indication of the need for more restricted guidelines, in order to prevent the evolution of this pathogen. Furthermore, new alternative therapies are urgently needed, due to the wide spreading of MDR phenotypes, which increases the possibility of treatment failure. We found that the human isolates exhibited high resistance patterns; meanwhile, the animal isolates were more virulent. Additionally, high MAR indices were recorded to the prescribed antibiotics within each hospital in this study. These findings were consistent with the hypothesis that the acquisition of antimicrobial resistance is always associated with the loss of bacterial fitness, and the prescribed antibiotics were consistently associated with the microbial resistance patterns. Finally, our results presented an indication of the diversity of *E. coli* isolates, but we did not support the possibility of its zoonotic transmission. Therefore, further investigations are needed to gather more information about the epidemiological aspects of DEC isolates, to identify the potential sources of their infections and to implement control measures for preventing their further spread.

## Figures and Tables

**Figure 1 antibiotics-11-00552-f001:**
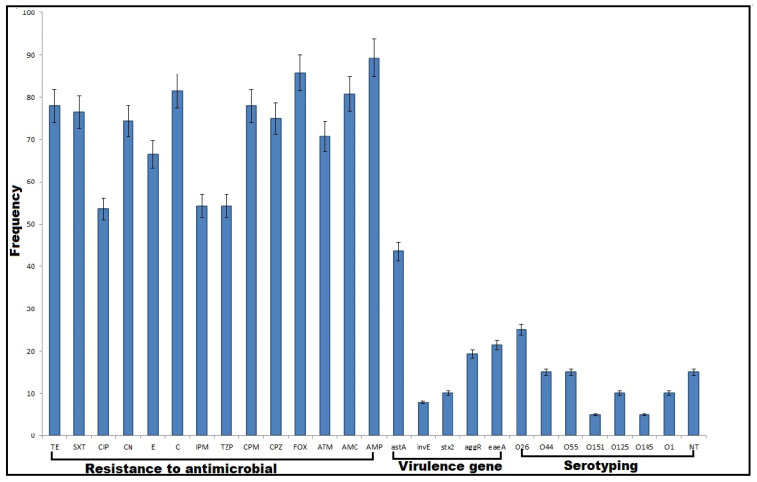
Percentages of antimicrobial resistance, virulence genes and serotypes in diarrheagenic *E. coli* isolates. AMP: ampicillin, CPZ: cefoperazone, FOX: cefoxitin, AMC: amoxicillin/clavulanic acid, ATM: aztreonam, CPM: cefepime, TZP: piperacillin/tazobactam, IPM: imipenem, C: chloramphenicol, E: erythromycin, CN: gentamycin, CIP: ciprofloxacin, SXT: sulfamethoxazole/trimethoprim, and TE: tetracycline. Genes encoding for adhesion (*fim*H and *kps*M), enterotoxin (*ast*A), intimin (*eae*A), Vero toxins (*vtx),* Shiga toxins (*stx*), hemolysin (*hly)*, outer membrane protein (*omp*) and transcriptional activators for aggregative adherence fimbriae (*aggR*) and invasion (*invE*) genes.

**Figure 2 antibiotics-11-00552-f002:**
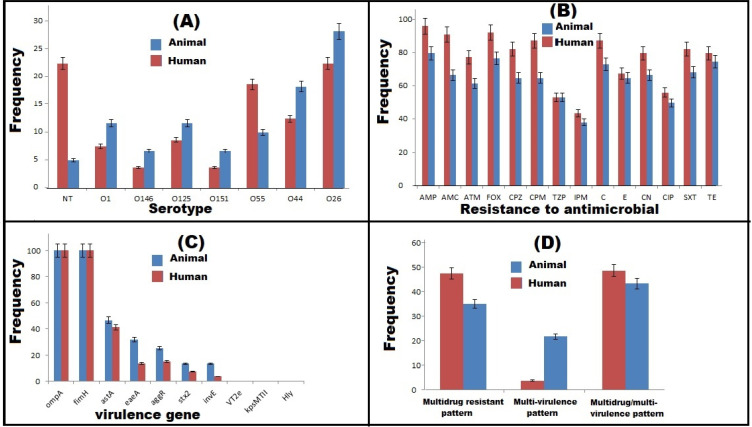
Distribution of serotypes (**A**), antimicrobial resistance (**B**), virulence genes (**C**) and multidrug resistance and multi-virulence patterns (**D**) among diarrheagenic *E. coli* isolates from human and animal sources. FOX: cefoxitin, AMP: ampicillin, ATM: aztreonam, AMC: amoxicillin/clavulanic acid, CPZ: cefoperazone, CPM: cefepime, TZP: piperacillin/tazobactam, IPM: imipenem, E: erythromycin, CIP: ciprofloxacin, C: chloramphenicol, SXT: sulfamethoxazole/trimethoprim, CN: gentamycin, and TE: tetracycline. Genes encoding for adhesion (*fim*H and *kps*M), enterotoxin (*ast*A), intimin (*eae*A), Vero toxins (*vtx),* Shiga toxins (*stx*), hemolysin (*hly)*, outer membrane protein (*omp*) and transcriptional activators for aggregative adherence fimbriae (*aggR*) and invasion (*invE*) genes.

**Figure 3 antibiotics-11-00552-f003:**
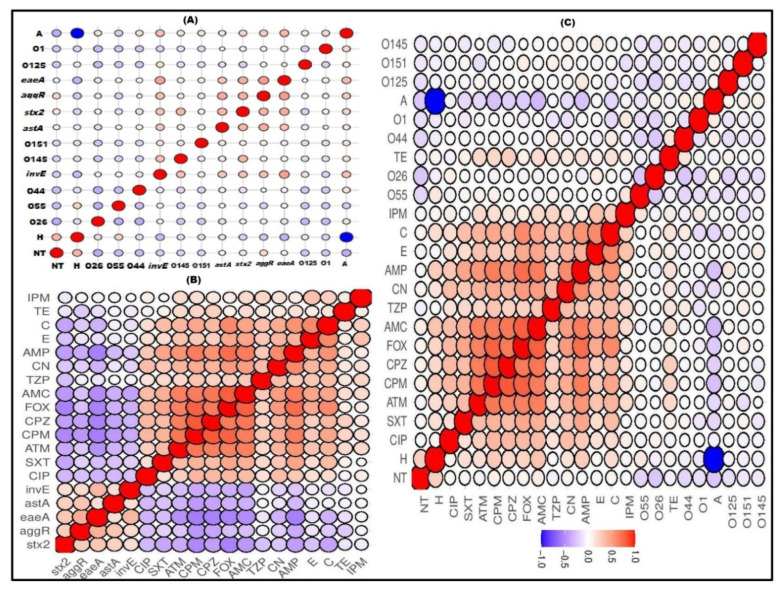
Correlation (R) between serotypes and virulence genes (**A**), virulence genes and antimicrobial resistance (**B**), and serotypes and antimicrobial resistance (**C**) of diarrheagenic *E. coli*. Blue and red colors indicate positive and negative correlation, respectively. The degree of correlation coefficient (R) is represented by the color key. The darker blue and red colors imply stronger negative (R = −0.5:−1) and positive (R = 0.5:1) correlations, respectively. Genes encoding for adhesion (*fim*H and *kps*M), enterotoxin (*ast*A), intimin (*eae*A), Vero toxins (*vtx),* Shiga toxins (*stx*), hemolysin (*hly)*, outer membrane protein (*omp*) and transcriptional activators for aggregative adherence fimbriae (*aggR*) and invasion (*invE*) genes. H; human, A; animal, AMP: ampicillin, AMC: amoxicillin/clavulanic acid, ATM: aztreonam, FOX: cefoxitin, CPZ: cefoperazone, CPM: cefepime, TZP: piperacillin/tazobactam, IPM: imipenem, C: chloramphenicol, E: erythromycin, CN: gentamycin, CIP: ciprofloxacin, SXT: sulfamethoxazole/trimethoprim, and TE: tetracycline.

**Figure 4 antibiotics-11-00552-f004:**
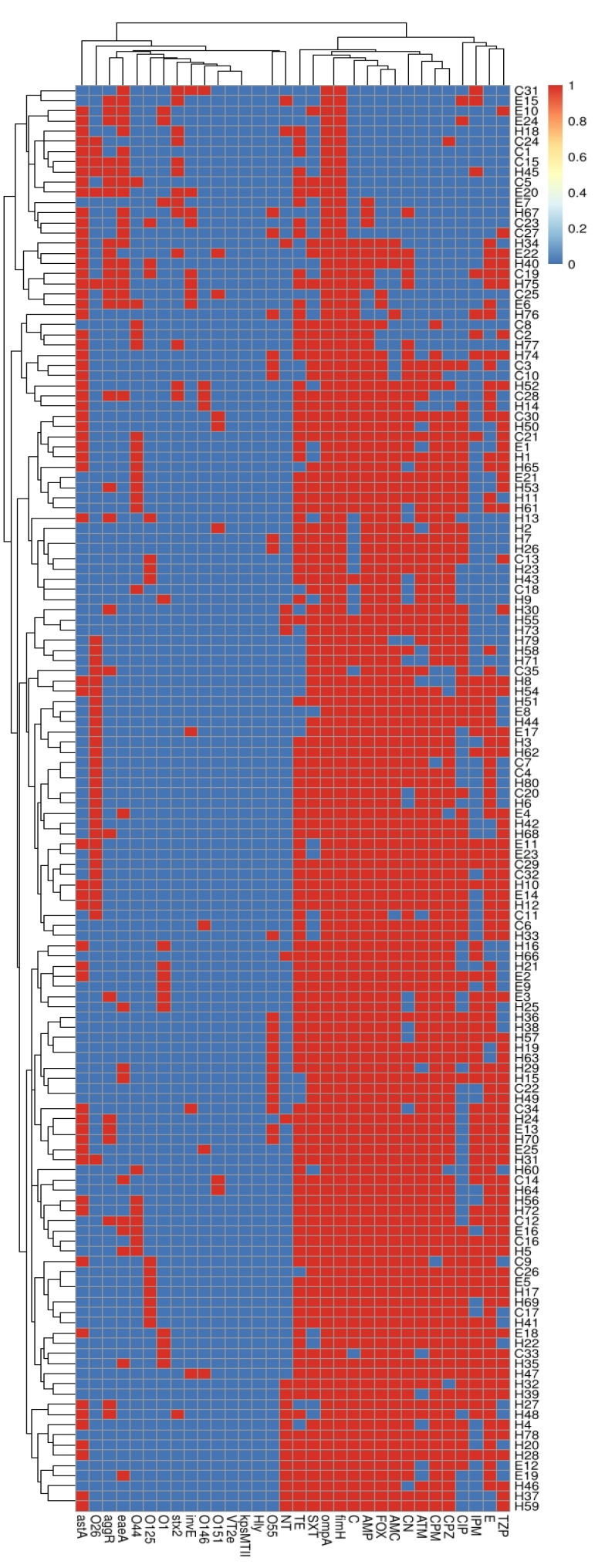
Heat map for the investigated diarrheagenic *E. coli* isolates, depending on the existence of virulence genes, antimicrobial resistance, and serotypes. The blue and red colors represent the sensitivity and resistance to particular antimicrobial drugs, and the absence and presence of certain serotype and virulence genes, respectively. The DEC isolates from human (H), cow (C) and equine (E) sources are coded on the right of the heat map. AMP: ampicillin, CPZ: cefoperazone, AMC: amoxicillin/clavulanic acid, CPM: cefepime, ATM: aztreonam, E: erythromycin, FOX: cefoxitin, TZP: piperacillin/tazobactam, IPM: imipenem, C: chloramphenicol, CN: gentamycin, CIP: ciprofloxacin, SXT: sulfamethoxazole/trimethoprim, and TE: tetracycline.

**Table 1 antibiotics-11-00552-t001:** Primer sequences and amplicon sizes of the corresponding virulence genes of DEC.

Target Virulence	Primers Sequences (5′-3′)	Amplicon Size (bp)	Reference
*omp*A	F: AGCTATCGCGATTGCAGTG	919	[59]
	R: GGTGTTGCCAGTAACCGG		
*kps*MTII	F: CAGGTAGCGTCGAACTGTA	280	[59]
	R: CATCCAGACGATAAGCATGAGCA		
*hly*	F: AACAAGGATAAGCACTGTTCTGGCT	1177	[60]
	R: ACCATATAAGCGGTCATTCCCGTCA		
*stx*2	F: CCATGACAACGGACAGCAGTT	779	[61]
	R: CCTGTCAACTGAGCAGCACTTTG		
*fim*H	F: TGCAGAACGGATAAGCCGTGG	508	[62]
	R: GCAGTCACCTGCCCTCCGGTA		
*vt2e*	F: CCT TAA CTA AAA GGA ATA TA	230	[63]
	R: CTG GTG GTG TAT GAT TAA TA		
*ast*A	F: TGCCATCAACACAGTATATCC	116	[64]
	R: TCAGGTCGCGAGTGACGGC		
*inv*E	F: CGATAGATGGCGAGAAATTATATCCCG	766	[65]
	R:CGATCAAGAATCCCTAACAGAAGAATCAC		

## Data Availability

All data in this study are presented in the submitted manuscript, and in the Appendix A.

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
