# Peer review of "Comparative Analysis of Human and Animal E. coli: Serotyping, Antimicrobial Resistance, and Virulence Gene Profiling"

_antibiotics, 2022, doi:10.3390/antibiotics11050552_

Round 1

Reviewer 1 Report

The manuscript entitled "Comparative analysis of human and animal E. coli : serotyping, antibiogram, and virulence genes profiling" is interesting, but needs significant improvement for publication. 

It will be good if you can explain the novelty and significance of the work specifically in the discussion section. The error bars are missing in the figures. Please provide a figure of better resolution for Figure 3. As it is a heat map representation, there is no need to display the individual values in the heatmap. Provide the correlation matrix values in the supplementary section.

Author Response

We would like to thank the reviewer for his raised and thorough comments and we go precisely through these comments to meet your expectations.

1-It will be good if you can explain the novelty and significance of the work specifically in the discussion section.

Thank you for this comment. In accordance with the reviewer comment, more information was introduced in the discussion section

2-The error bars are missing in the figures.

Thank you for this comment; It was added to the figure

3-Please provide a figure of better resolution for Figure 3. As it is a heat map representation, there is no need to display the individual values in the heatmap. Provide the correlation matrix values in the supplementary section

Thank you for good reviewing. Therefore, all these points were managed in accordance to this comment and a correlation matrix was added as a supplementary section.

Reviewer 2 Report

  • Please provide a full description of abbreviations of the virulence factors for Figure 1. and Figure 2.
  • Please indicate the full description of the word MAR at its first appearence in line 175.

Author Response

We would like to thank the reviewer for his raised and thorough comments and we go precisely through these comments to meet your expectations.

1-Please provide a full description of abbreviations of the virulence factors for Figure 1. and Figure 2.

Thank you for this comment. All abbreviations were added

2-Please indicate the full description of the word MAR at its first appearence in line 175.

Thank you for this comment. This was managed in accordance to the reviewer recommendation

Reviewer 3 Report

General comment

This manuscript was focused on characterization of E. coli strains isolated from humans and animals in an Egypt area. It provide some genetic features of these strains in order to evaluate the risk associated for the public health. However, this aspect is not emerged from the text and many drawbacks were present for introduction, methodology and conclusions were not so elucidated. As reported in the following section dedicated to specific comments, important lack were identified. Hence, this work cannot be accepted for publication.

Specific comments

Introduction

Introduction is too limited. Authors should enrich this section adding more and more information about the spread of E. coli in different niches, the spread of antibiotic-resistance strains and a better description of illnesses and symptoms to clarify the significance of this study for the public health.

Title should be contain “antibiotic-resistance” instead “antibiogram”

Keywords: my suggestion is to replace “heterogeneous nature” with more specific keyword such as “genetic diversity”

Line 46: Escherichia coli must be in italic throughout the text

From line 67 to 75. This paragraph needs to be reframed because the list of virulence factors is not clear due to the misuse of “;”. Please, list virulence factors previously described and put “;” only after the abbreviation of factors.

Line 71. Remove a space after between or and stx and check all spaces throughout the text

Line 81. Introduce a space after dot

Results

Line 90 and 91: this aspect is not clear. What it means “The results of the API 20E system and PCR amplification of the 16SrRNA gene were in agreement with those of conventional methods”? What are the conventional methods mentioned by authors?

Figure 1 is not necessary if authors report the same information in figure 2 where distribution of antimicrobial resistance, virulence and serotyping were described highlighting them among human and animals. My suggestion is to provide only figure 2 that is more complete.

Figure 3 is not readable. Authors should replace it choosing a different method to show these results.

Materials and methods

Paragraph 4.1 was focused on isolation and identification of DEC, but some details were not reported. For example, no incubation temperature has been indicated as well as the period of incubation. Authors should report a brief description of PCR conditions because from the mentioned reference [37] is not so immediately possible obtain the protocol.

Paragraph 4.2 was limited to the description of guidelines of manufacturer’s recommendations.

From line 333 to line 335. Was MIC performed for all strains or only for those whose results were unclear? Why didn't the authors make the mic right away?

Paragraph 4.4 Authors should better describe MAR index. For example adding that “A MAR greater than 0.2 means that the high risk source of contamination is where antibiotics are frequently used (Rotchell D, Paul D. Multiple Antibiotic Resistance Index. Fitness and Virulence Potential in Respiratory Pseudomonas aeruginosa from Jamaica. Journal of Medical Microbiology. 2016;65:251–271).

Line 357. Authors should use the abbreviation DEC as first mentioned throughout the text

Discussion

The first section is a description of other species and it was not dedicated to E. coli. My suggestion is remove this part.

Authors talk about correlations in many parts but these correlations were not clearly identified or analysed. This is an important aspect because from these correlations the emergence of significant pathogenicity of strains could be obtained.

Conclusions

Line 387 how do the authors talk about prescribed antibiotics? No clinical analysis of the human or animal was done.

In general, this section is not consistent with results and it doesn’t provide significant conclusion for the work.

Author Response

We would like to thank the reviewer for his raised and thorough comments and we go precisely through these comments to meet your expectations.

Introduction

1-Introduction is too limited. Authors should enrich this section adding more and more information about the spread of E. coli in different niches, the spread of antibiotic-resistance strains and a better description of illnesses and symptoms to clarify the significance of this study for the public health.

Thank you for this comment. More information was added in the introduction section in accordance to the reviewer comment

2-Title should be contain “antibiotic-resistance” instead “antibiogram”

Thank you for this comment. It was modified in the new version of our manuscript

3-Keywords: my suggestion is to replace “heterogeneous nature” with more specific keyword such as “genetic diversity”

Thank you for this comment. It was replaced in the new version of our manuscript

4-Line 46: Escherichia coli must be in italic throughout the text

Thank you for this comment. It was revised throughout the text and adjusted

5-From line 67 to 75. This paragraph needs to be reframed because the list of virulence factors is not clear due to the misuse of “;”. Please, list virulence factors previously described and put “;” only after the abbreviation of factors.

Thank you for this comment. This paragraph was reframed

6-Line 71. Remove a space after between or and stx and check all spaces throughout the text

Thank you for this comment. It was removed and it was revised throughout the manuscript

7-Line 81. Introduce a space after dot

Thank you for this comment. It was added

Results

1-Line 90 and 91: this aspect is not clear. What it means “The results of the API 20E system and PCR amplification of the 16SrRNA gene were in agreement with those of conventional methods”? What are the conventional methods mentioned by authors?

Thank you for this comment. The conventional methods (cultural, morphological and biochemical characteristics) were elucidated in the material and methods section in the revised version. We mean that all DEC isolates, which were detected by cultural, morphological and biochemical characteristics, were confirmed with API 20E and genetic detection of 16SrRNA gene without false results in the conventional methods

2-Figure 1 is not necessary if authors report the same information in figure 2 where distribution of antimicrobial resistance, virulence and serotyping were described highlighting them among human and animals. My suggestion is to provide only figure 2 that is more complete.

Thank you for this comment. Figure 1 describes the frequency of antimicrobial resistance, virulence and serotyping data for all isolates; meanwhile, Figure 2 highlights these criteria for both human and animal isolates separately. Therefore, the data in the two Figures are completely different

3-Figure 3 is not readable. Authors should replace it choosing a different method to show these results.

In accordance with your comment and the other reviewers points of view, the resolution of Figure 3 was adjusted and more information was added in the main text and in the supplementary data

Materials and methods

1-Paragraph 4.1 was focused on isolation and identification of DEC, but some details were not reported. For example, no incubation temperature has been indicated as well as the period of incubation. Authors should report a brief description of PCR conditions because from the mentioned reference [37] is not so immediately possible obtain the protocol.

Thank you for this comment. More information was added

2-Paragraph 4.2 was limited to the description of guidelines of manufacturer’s recommendations.

Thank you for this comment. The full methodology for this test was added

3-From line 333 to line 335. Was MIC performed for all strains or only for those whose results were unclear? Why didn't the authors make the mic right away?

Thank you for this comment. The antimicrobial susceptibility of all DEC isolates was done using both disc susceptibility testing and broth microdilution methods and this was elucidated in the main text.  Disc sensitivity was used as a preliminary test and the broth microdilution was used as a confirmatory test 

4-Paragraph 4.4 Authors should better describe MAR index. For example adding that “A MAR greater than 0.2 means that the high risk source of contamination is where antibiotics are frequently used (Rotchell D, Paul D. Multiple Antibiotic Resistance Index. Fitness and Virulence Potential in Respiratory Pseudomonas aeruginosa from Jamaica. Journal of Medical Microbiology. 2016;65:251–271).

Thank you for this comment. It was rephrased and more information was added

5-Line 357. Authors should use the abbreviation DEC as first mentioned throughout the text

Thank you for the good reviewing, it was corrected

Discussion

1-The first section is a description of other species and it was not dedicated to E. coli. My suggestion is remove this part.

Thank you for this comment. It was removed and additional information was added according to other reviewers' comments

2-Authors talk about correlations in many parts but these correlations were not clearly identified or analysed. This is an important aspect because from these correlations the emergence of significant pathogenicity of strains could be obtained.

Thank you for this comment. All correlations were revised and more information in this aspect was added

Conclusions

1-Line 387 how do the authors talk about prescribed antibiotics? No clinical analysis of the human or animal was done.

Thank you for this comment. Our isolates were clinical, where they were isolated from different patients within university and private hospitals in two different Governorates. The full data of the prescribed antibiotics were available. As it was expected, we found a strong positive correlation between the prescribed antibiotics and the antimicrobial resistant patterns of our isolates as elucidated in the discussion and conclusion sections.  

2-In general, this section is not consistent with results and it doesn’t provide significant conclusion for the work.

Thank you for this comment. The conclusion section was rephrased and modified in accordance with the reviewer comment

Round 2

Reviewer 3 Report

Authors provided sufficient correction and the manuscript has been improved. Hence, it can be accepted in this version.